# Nutritional Status, Dietary Intake and Dietary Diversity of Landfill Waste Pickers

**DOI:** 10.3390/nu14061172

**Published:** 2022-03-10

**Authors:** Elizabeth C. Swart, Maria van der Merwe, Joy Williams, Frederick Blaauw, Jacoba M. M. Viljoen, Catherina J. Schenck

**Affiliations:** 1Department of Dietetics and Nutrition, Faculty of Community and Health Sciences, University of the Western Cape, Bellville, Cape Town 7530, South Africa; joycyster08@gmail.com; 2School of Public Health, University of the Western Cape, Bellville, Cape Town 7530, South Africa; smvandermerwe@gmail.com; 3School of Economic Sciences, North-West University, Potchefstroom Campus, Private Bag X6001, Potchefstroom 2520, South Africa; derick.blaauw@nwu.ac.za; 4School of Economics and Econometrics, University of Johannesburg, Johannesburg 2092, South Africa; kotiev@uj.ac.za; 5Department of Social Work, DSI/NRF/CSIR Chair in Waste and Society, University of Western Cape, Bellville, Cape Town 7530, South Africa; cschenck@uwc.ac.za

**Keywords:** waste pickers, nutritional status, dietary intake, dietary diversity, South Africa

## Abstract

The purpose of this study was to investigate and describe the nutritional status, dietary intake and dietary diversity of waste pickers in South Africa, a socioeconomically vulnerable group who makes a significant contribution to planetary health through salvaging recyclable material from dumpsites. Participants were weighed and measured to calculate body mass index (BMI). Dietary intake was recorded using a standardised multipass 24 h recall. Individual dietary diversity scores were derived from the dietary recall data. Data were collected from nine purposefully selected landfill sites located in six rural towns and three cities in four of the nine provinces in South Africa, providing nutritional status information on 386 participants and dietary intake on 358 participants after data cleaning and coding. The mean BMI of the study sample was 23.22 kg/m^2^. Underweight was more prevalent among males (22.52%) whilst 56.1% of the females were overweight or obese. The average individual dietary diversity score was 2.46, with 50% scoring 2 or less. Dietary intake patterns were characterised as monotonous, starch-based and lacking vegetables and fruits. The nutritional status, dietary intake and dietary diversity of waste pickers reflect their precarious economic status, highlighting the need for health, social and economic policies to improve access and affordability of nutritious food.

## 1. Introduction

Nutrition is a critical component of health and development, with adequate nutrition aiding the prevention of malnutrition in all its forms, improving productivity and creating opportunities to gradually break the cycles of hunger and poverty.

South Africa is an upper-middle-income country with a culturally diverse population estimated at 59.6 million people [1]. Social, health and economic disparities remain and are aggravated by persistent high poverty, entrenched structural inequality and unemployment [2,3,4]. The country is experiencing rapid epidemiological transition, with noncommunicable diseases (NCDs) increasing to be the major cause of death, with an estimated 269,000 NCD related deaths annually [5]. This increase as well as the substantial higher burden is particularly noted among lower socioeconomic groups and obese persons [6,7,8].

The nutrition situation in South Africa is complex and typical of a country in nutrition transition. Undernutrition, notably stunting and micronutrient deficiencies, coexist with a rising incidence of overweight and obesity and the associated consequences such as NCDs [9]. The South African food system is highly commercialised with the majority of households purchasing all their food. The food system is furthermore characterised by cultural and socioeconomic diversity and high levels of income inequality, rendering vulnerable population groups at risk of food insecurity and hunger [10].

The percentage of the population that experienced hunger decreased from 29.3% in 2002 to 11.3% in 2018 and the percentage of people with limited general access to food decreased from 29.1% to 23.8% from 2010 to 2018 [10]. However, the COVID-19 pandemic foregrounded the failures of the food system to provide sufficient, healthy, nutritious food and to serve the most vulnerable people in South Africa, potentially reversing progress achieved in the reduction of hunger of the past two decades. The National Income Dynamics Survey, Coronavirus Rapid Mobile Survey (NIDS-CRAM), a monthly nationally representative panel survey conducted since May 2020, indicated immediate adverse effects on employment and food security and widening inequality. Between March and June 2020, 40% of the NIDS-CRAM sample reported the loss of employment as a result of COVID-19 and 22% of adults and 15% of children were reported to have gone to bed hungry [11,12].

Waste is an unavoidable byproduct of human activities and if accumulated in large quantities, it can lead to degradation of the environment, natural resources and health problems [13]. It is estimated that South Africa generated approximately 54.2 million tons of waste in 2017, of which an estimated 20.7 million tons were recycled. In addition, South Africa produces approximately 31 million tons of food annually, of which an estimated 10 million tons (34.3%) is lost to waste [14].

The commercialisation of waste management, coupled with an increase in recycling practices and unabated unemployment resulted in the survivalist activity of informal waste picking by the poorest and marginalised on landfill sites [15,16]. Their diversification of livelihood is an important strategy towards poverty alleviation. Waste pickers collect, sort, recycle, repurpose and/or sell materials thrown away by others; salvaging and revaluing waste through the recycling value chain [17]. While some rummage in search of necessities, others collect and sell recyclables to middlemen or businesses or work in recycling warehouses or plants owned by their cooperatives or associations [15,18]. In South Africa, there are an estimated 60,000 to 90,000 informal waste pickers operating at different levels [18].

These informal waste pickers operate at the lowest level in the hierarchy of entities that collect and dispose of waste, including municipal waste collection services and recycling structures. Yet they significantly contribute to the economy in general and the recycling economy in particular [19]. Nationally, it is estimated that waste pickers save municipalities up to R750 million per year by collecting waste at no cost and saving landfill space [20]. While waste pickers are typically scorned and treated as a nuisance, their valuable contribution towards waste management and recycling industries are increasingly being recognised and valued by the South African government and stakeholders in the waste industry; with several attempts to integrate waste pickers into formal waste and recycling structures [19]. Their contribution to the economy is in stark contrast to the meagre, and extremely variable, mean financial benefit from informal waste picking trade activities of R451.90 (less than 30 USD) per week in a case study by Schenck et al. in 2018 [21]. Considering that each waste picker in the above case study had a mean of four financial dependents, their household income would seldom exceed the poverty line of R1 268 (82 USD) per person per month [22].

Waste pickers were particularly hard hit by the initial national lockdown regulations following the outbreak of COVID-19, demonstrating their vulnerability to socioeconomic factors. As an informal sector, waste picking was not regarded as an essential service and waste pickers were dismissed for the role they play in the waste sector, resulting in income loss, hunger and reliance on food relief efforts. Reports regarding a lack of consultation and communication with stakeholders demonstrate how the impact of the pandemic extended beyond the immediate risk of infection to further adverse effects related to policy decisions [23].

Although it has been documented that waste pickers obtain food through their activities, the exact contribution of their “findings” to their dietary intake is difficult to quantify, and largely unknown. No scientifically analysed information is available on the nutritional status and dietary intake of waste pickers in South Africa. The objectives of the study were to (1) assess the nutritional status of people who make a living of informal waste picking from landfill sites, (2) to assess the dietary intake of waster pickers, and (3) to assess the dietary diversity of waste pickers.

## 2. Materials and Methods

This study on the nutritional status, dietary intake and dietary diversity of waste pickers on landfill sites in South Africa, collected quantitative cross-sectional data from nine purposefully selected landfill sites (in six rural towns and three cities and representing both outsourced and municipally managed landfill sites) in four of the nine provinces in South Africa in 2015.

Trained fieldworkers, proficient in the languages spoken on the respective landfill sites, collected socioeconomic data using a questionnaire with quantitative and open-ended explanatory, qualitative questions where more information was required. Dietary intake was recorded by the trained field workers, using a standardised 24 h recall record form and a standardised dietary intake toolkit to assist with quantification [24]. Where possible, interview days were selected to ensure that 24 h recalls represented week and weekend days proportionally. Measurements of weight and height were obtained by one researcher, to limit interobserver variability. Bodyweight was recorded to the nearest 0.1 kg (A&D Personal Precision Scale, Tokyo, Japan) and height to the nearest 1mm (using a portable stadiometer), applying standardised methodology and were used to calculate body mass index (BMI) according to World Health Organization (WHO) criteria [25].

All waste pickers present on the landfill site on the day of data collection who were willing to participate were interviewed. A total of 409 adult waste pickers were interviewed and missing values were recorded for some participants. Discrepancies between numbers of waste pickers with dietary intake, anthropometric measurements and general sociodemographic information are the result of waste pickers’ choice not to continue due to time constraints when new trucks with fresh waste arrived. No missing values were recorded for the two smallest landfill sites. Statistical analyses of selected variables in each data set revealed no systematic bias for those participants with missing values (independent *t*-test all *p*-values were ≥0.05; paired sample *t*-test after replacing missing values with series mean for continuous variables could not be performed as the standard error of the difference was 0) See Appendix A. Subsequently, information on all participants was included in the analyses using a pairwise exclusion analysis by analysis when applicable.

The study was approved by the Senate Research Ethics Committee of the University of the Western Cape (reference number 15/4/24). After receiving information on the process and purpose of the research, participants completed written consent to confirm voluntary participation. Illiterate participants gave verbal consent in the presence of a third person as a witness. Consent forms and information sheets were translated in the most spoken languages and explained in the language best understood. Participants were compensated for their involvement in the form of food worth 1 USD, following the interviews and anthropometric measurements. Questionnaires and data collection forms were anonymised and identified by a code for data analysis purposes. All information acquired was stored securely. Hard copies were stored in a sealed box in a locked cabinet and electronic data was stored on a password-protected computer.

### Data Analysis

Anonymised data, recorded in Microsoft Excel 2016, was exported and analysed in IBM SPSS Statistics 25, 2017 for descriptive analysis. Pairwise exclusion was performed and cases were excluded if the variables under investigation were missing.

Anthropometric data was recoded and analysed according to BMI categories. The anthropometric status of male and female waste pickers was compared to the findings of the South African Demographic Health Survey (SADHS) of 2016 reported in their key findings report [26]. The 2016 SADHS used a stratified, two-stage sampling design to provide estimates that were representative of national, provincial and locality type (urban and nonurban areas) and included a sample of 11,083 households. Both studies used the same BMI categories for underweight, overweight and obesity and excluded pregnant women and women who have given birth in the last two months.

The dietary intake data collection was done according to the Centre of Excellence in Food Security Manual on Dietary intake assessment—24 h recall [24]. Quantified 24 h dietary recall data were coded and converted to nutrient intakes using the electronic South African Medical Research Council Food Composition Database. Acceptable macronutrient distribution ranges (AMDR) were used to assess macronutrient intake of participants [27]. Accordingly, the AMDR for carbohydrates is 45–65% of energy, for protein it is 10–35% of energy and for fat it is 20–35% of energy.

Micronutrient intake of participants were compared to the dietary reference intake (DRI) values for men and women aged ≥ 19 years [28]. The estimated average requirements (EARs) according to gender and age groups were applied, when established, and adequate intakes (AIs) when EARs were not established.

Dietary adequacy was determined through individual dietary diversity scores. The 24 h recall data was cleaned and coded according to each of sixteen food groups, namely: cereals; white roots and tubers; vitamin A-rich vegetables and tubers; dark green leafy vegetables; other vegetables; vitamin A-rich fruits; other fruits; organ meat; flesh meats; eggs; fish and seafood; legumes, nuts and seeds; milk and milk products; oils and fats; sweets; and spices, condiments and beverages. A dietary diversity score was calculated by summing the number of nine aggregated food groups (starchy staples; dark green leafy vegetables; vitamin A-rich fruits and vegetables; other fruits and vegetables; organ meat; meat and fish; eggs; legumes, nuts and seeds; and milk and milk products) from which foods had been consumed, with a potential score range of 0 to 9 [29]. Only foods consumed in quantities above 1 tablespoon (15 g) were included and each group was only counted once.

## 3. Results

### 3.1. Sociodemographic Characteristics

The majority (234) of the 409 participants were male, with six of the sites being male dominated, one site allowed only male waste pickers and the remaining two sites had predominantly female waste pickers. While some of the participants did not know their age, the mean age of the 88.3% of the respondents who knew their age was 39 years. Female participants were on average 5 years older than their male counterparts. The majority (82.1%) of the study participants were black Africans and the remainder was of mixed ancestry (Coloured South Africans). A small proportion (7.9%) of the participants completed secondary education and none had any tertiary education. The income of waste pickers varied greatly depending on the types of waste delivered to landfill sites. This is described elsewhere [30,31]

### 3.2. Nutritional Status

Of the 409 participants, anthropometric measurements were taken for 386 individuals. The mean BMI of males and females were 20.9 ± 2.9 kg/m^2^ and 26.4 ± 6.3 kg/m^2^ respectively, with most males (74.9%) having a normal BMI, compared to 42.4% of the females. While underweight was more prevalent among males (16.4%) than females (6.7%) (see Appendix A), 50.9% of the females were overweight or obese compared to 8.7% of the males (Table 1).

### 3.3. Dietary Intake

#### 3.3.1. Energy Intake

The mean daily energy intake, further described in Table 2, was 5664.1 kJ for female participants and 7408.3 kJ for males. These intakes were below the DRI and met 75% and 80% of the daily requirements respectively. The daily energy intake reported by participants varied from 110.4 kJ to 26,147 kJ.

#### 3.3.2. Macronutrient Intake and Distribution

The proportional adequacy of macronutrient and fibre intake is summarised in Table 2. The mean carbohydrate intake was 248.6 g/day for males and 210.6 g/day for females, contributing 191% and 162% of the minimum recommended intake respectively. The mean carbohydrate intake of the majority (51.6%) of participants contributed more than 65% of the total daily energy intake. A higher percentage of women (63.2%) consumed carbohydrates at levels above the acceptable macronutrient distribution range (AMDR), compared to 42.7% of men. Overall, the daily carbohydrate intake varied from 6.0 g to 961.9 g per participant. Carbohydrates were mostly consumed as foods from the cereal and white roots and tubers groups, while 69.5% of the participants also consumed foods from the sweets group (Appendix A) which includes sugar, sugar-sweetened beverages, sweetened fruit juice and sugary foods such as candies, cookies and cake. The mean intake of added sugar was 40.7 g overall and 44.1 g and 36.0 g for males and females respectively with 42.2% consuming more than 25 g per day. More men (26.9%) had an excessive intake of added sugar compared to women (15.3%). Overall, 38.5% of the participants consumed sugar-sweetened beverages, with intake ranging from 2 to 7 L per person amongst those individuals, while only 0.6% consumed fruit juice. A small proportion of the participants (7.2%) reported having consumed alcohol in the recall period, which contributed to their total carbohydrate intake.

The mean total protein intake for women was 43.5 g/day and met 93% of the DRI protein requirement. In addition, the mean total protein intake for men met 109% of the requirement, at 61.5 g/day. However, 23.5% of the participants reported protein intake below the AMDR (Table 2). The total daily protein consumption varied from 0g to 203.8 g per participant for the recalled day. Overall, 61.5% of the participants consumed flesh meat during the 24 h recall period, 8% consumed fish or seafood, 5.5% consumed legumes, seeds and nuts and 4.4% consumed organ meats. Processed meats were consumed by 14.4% of the participants (Appendix A).

Very few (16%) of the participants reported fat intake at levels exceeding the recommended macronutrient distribution and half of the study sample (50%) consumed fat at levels below the AMDR. Fat intake per person varied from 0 g to 225.6 g for the 24 h recall.

Only half (50%) of the participants consumed oils and fats, used for cooking, during the reporting period. Amongst the 13% of the study sample with an excessive fat intake, this was attributed to the consumption of foods with high fat content, including fried chicken, vetkoek, chicken feet, Russian sausages (the processed meat product), maas (fermented dairy product), peanut butter, beef with fat and processed beef patties.

#### 3.3.3. Micronutrient Intake

The mean intake of micronutrients was mostly above 80% of the DRI for both male and female participants, as summarised in Table 3. However, the intake of calcium, vitamin E and pantothenic acid was below 80% of the DRI; and very low (below 30% of DRI) for potassium, vitamin C and vitamin D. The DRI for iron in females varies according to age, due to increased iron needs during childbearing years. The intake for all females were analysed collectively and not according to their age, but the mean iron intake for women was above the recommended intake for all age groups. The DRIs for biotin was not met for females and males, but the mean intake level of these micronutrients was above 80% of the DRI.

The mean sodium intake was 58.9% of the DRI for women and 97.6% of the DRI for males. The salt intake of 16% of the study sample, who reported excessive sodium intake, was checked and included farm-style sausages and Russian sausages (processed meat products), maas (fermented dairy product) and sugary drinks. No added sodium intake was reported in the 24 h recall data and sodium was therefore mostly consumed through processed foods.

### 3.4. Dietary Diversity

The average individual dietary diversity scores were low and did not vary greatly between sites (Figure 1 and Appendix A). A low dietary diversity score is associated with food insecurity as well as a risk of micronutrient deficiency. Very low consumption of foods from the fruit and vegetable groups, particularly vitamin A-rich fruits and vegetables, was reported. Less than a quarter (23.8%) of the participants consumed milk and milk products, while only 5.5% of the participants consumed legumes, nuts and seeds. The types of food reported to be consumed by waste pickers are summarised in Table 4, according to food groups.

## 4. Discussion

While dietary intake of the study participants varied greatly, the total energy consumption was below the recommended intake range. With regards to macronutrient distribution, most participants had an excessive consumption of carbohydrates, adequate intake of proteins and inadequate intake of fats. None of the participants met the recommended intake for fibre. Micronutrient intake was mostly adequate according to the recommended dietary intake, except for potassium, vitamin C and vitamin D. While the sodium intake for the study population was at an acceptable level, a proportion of the population had a very high sodium intake due to the type of foods found at the sites where they worked. The overall dietary diversity among study participants was very low, with little variation between sites.

Consumption of a variety of foods is required to ensure adequate nutrient intake. The availability and affordability of highly processed foods are considered important drivers of poor nutrition [32]. On the other hand, dietary patterns characterised by higher intakes of unprocessed foods are linked to more positive health outcomes [33]. However, for the most vulnerable groups, nutrient-rich foods such as animal-source foods, fruits and vegetables are not affordable [34,35].

The quarterly labour force survey of the fourth quarter of 2020 reports the official unemployment rate in South Africa at 32.5% [36]. Many individuals and households, therefore, rely on social support as a source of income, with 31% and 44.3% respectively being dependent on social grants in 2018 [10]. Thirty-four per cent of participants benefitted from social grants with either themselves or other household members receiving a grant [30].

Waste picking is not a choice or preference, but rather a source of income for those desperate to make a living, and in some instances also a source of food. People with a low income may consume a less healthy diet due to energy-dense foods being relatively cheap sources of energy, but with a low nutrient density. A healthy diet remains unaffordable for most South Africans [34,35]. Waste pickers are regarded as vulnerable to economic instability. A high level of food insecurity—not having physical and/or economic access to sufficient food to meet dietary needs for a productive and healthy life at all times—was measured among the waste pickers included in this study and reported elsewhere [21]. Overall, 20% of the participants reported going to bed hungry at night and a further 18% reported going for a whole day and night without eating. In this study, two participants reported not eating or drinking anything during the previous day. Food sources included food brought from home, bought as ready meals, found on the landfill site or shared by other waste pickers [21].

The morbidity and mortality rates related to obesity and NCDs are higher among people from socioeconomically disadvantaged groups, due to poor food choices which mostly include energy-dense foods and low nutrition density [37]. The prevalence of obesity among the adult population in South Africa is increasing alarmingly, with more than a quarter of the female adult population estimated to be overweight and almost a third obese [38]. The prevalence of overweight and obesity in women has increased by 21% between 1998 and 2016 [9,38]. In addition, 31% of adult men are overweight or obese, an increase of 2% since 1998 [9,38]. In contrast to the burden of overweight and obesity, 3% of the South African female adult population and 10% of the male adult population are underweight and a dual burden of child undernutrition and adult obesity exists with 27% of children under the age of 5 years being stunted—a sign of chronic undernutrition [38].

The BMI of more than half of the waste pickers in this study was within the normal range. Underweight and overweight/obesity of female and male waste pickers were statistically significantly different. Of concern, the prevalence of underweight among male participants was double that of the national male population [38]. As illustrated in Table 1, underweight in both males and females were higher than the lowest wealth quantile in the national SADHS, 2016 [25]. Overweight and obesity was low in males, but in the female study population it was comparable to the lowest wealth income quantile of the SADHS 2016 albeit lower compared to the general South African population [38]. It is known that the prevalence of overweight and obesity varies among different population groups and the severity of obesity increases with increasing wealth [38], although overweight and obesity is very high in South African females regardless of wealth quantile. The lower prevalence of overweight and obesity among the study population, compared to the national population is therefore potentially attributed to their socioeconomic status. The prevalence of overweight and obesity however remains a concern, due to the association with NCDs, the leading cause of death in South Africa [2]. Access to public health services for NCD related treatment for waste pickers may be limited especially given the high opportunity cost as their daily survival is based on their engagement at the landfill sites at critical times of waste deposits.

Representing the leading cause of mortality worldwide (71% of all deaths), the prevalence of NCD multimorbidity is exploding in low- and middle-income countries, and substantially increasing in sub-Saharan Africa, particularly among lower socioeconomic groups and obese persons [9,10,11]. A large study (*n* = 1025) amongst waste pickers in the largest dumpsite in Latin America found 32.6% to be overweight and 21.1% obese, with a 24.2% prevalence of hypertension 24.2% and 10.1% of diabetes [39]. Another study from Brazil established that 29.5% of 253 male-dominant (86.2%) pushcart waste pickers in the city of Santos were overweight or obese [40]. Auler et al. (2014) reported a much higher prevalence of overweight (51.1%) and obesity (25.7%) with hypertension (32.8%) and diabetes (11.4%) among 268 waste pickers in southern Brazil [41].

In the absence of national data on the dietary intake of adult South Africans, a recent review of dietary surveys in the adult South African population from 2000 to 2015 [42] concluded that energy intake varied from low intakes in informal settlements to very high intakes in urban centres. Macronutrient intake varied similarly to energy intakes but remain within acceptable minimum distribution ranges [42]. Overall, food consumption patterns in the country have dramatically shifted towards a concerning overall increase in daily energy intake, a diet of sugar-sweetened beverages, an increase in the proportion of processed foods and animal source foods, added sugar and a shift away from vegetables [43].

Many low-income households consume monotonous, low-quality diets, typically cereal-based and lacking in vegetables, fruit and animal-source foods [44]. Monotonous diets and a lack of dietary diversity is closely associated with food insecurity [45]. The risk of micronutrient deficiencies also increases when dietary diversity is low [46] although it should be noted that mandatory fortification of maize meal, wheat flour and bread [47] is assumed to have contributed to the micronutrient content of diets of the waste pickers. Dietary diversity is a proxy for nutrient adequacy [29]. National food diversity was evaluated in 2009 as a proxy for food security, with a dietary diversity score below 4 regarded to reflect poor dietary diversity and poor food security [48]. Overall, a national level dietary diversity score of 4.02 reflects that the majority of South Africans consumed a diet low in dietary variety, albeit with significant provincial differences. Low dietary diversity correlated with socioeconomic status, lower-income groups having the lowest dietary diversity score of 2.93 [48]. At 2.465, the average dietary diversity score of waste pickers included in this study is lower than the average score of low-income groups in the national evaluation; implying that waste pickers are at explicit risk of food insecurity and micronutrient deficiencies due to inadequate diets.

Food groups most commonly consumed, according to the evaluation of national dietary diversity [48], included starchy staples, meat and fish, dairy and vegetables other than vitamin A rich food. Eggs, legumes and vitamin A-rich fruits and vegetable were the least consumed among the national population. Similarly, the most commonly deficient food groups observed in a recent review of dietary surveys in South Africa included vegetables and fruits, and milk and milk products [42]. Based on reported intake over the past 24 h, 49% and 59% of participants in the 2016 Demographic and Health Survey reported consuming fruit and vegetables respectively, with variations between population groups and according to income status [38]. It is a global trend for low-income households to spend less on fresh produce than higher-income households [49]. Of further concern, the consumption of fresh vegetables in South Africa declined whilst consumption of ultra-processed foods increased dramatically between 1994 and 2012 [43].

Fruit and vegetable consumption amongst waste pickers in this study is much lower than the reported national consumption and confirms that these foods are not accessible and/or affordable to socially vulnerable populations. A low intake of fruits and vegetables are of particular concern in relation to micronutrient deficiencies. Meat was the food most commonly collected on landfill sites [21]. This would explain why most of the waste pickers consumed foods from the meat and fish group, contrary to what would be expected due to their socioeconomic status as it was sometimes obtained from the sites and not bought with household income. The variation in overall dietary intake among waste pickers in this study is likely an indication of varied access to food retrieved from the landfill site.

Frequent consumption of food products not beneficial to health, such as sugary drinks, is of concern. According to the 2016 South African Demographic Health Survey [38], 36% of participants reported drinking any sugar-sweetened beverage (607 mL on average) and 14% reported drinking fruit juice (304 mL on average). A similar proportion of waste pickers in this study reported consuming sweetened beverages. The absence of access to water on the landfill sites [30] may be contributing to the consumption of sugary beverages.

With food production, processing and marketing being driven by profit, ultraprocessed food is becoming increasingly available and affordable [50]. At the same time, food security and the nutritional status of the most vulnerable population groups are likely to deteriorate further due to the health and socioeconomic impacts of the COVID-19 pandemic [44]. In South Africa, it is estimated that 9.34 million people faced high levels of acute food insecurity by the end of 2020 with projections of a further increase with regards to the number of people affected.

Particular attention should be paid, through policy interventions, to curb increasing food prices and the energy cost of food preparation. Access to and the affordability of nutrient-dense foods, by the most vulnerable and marginalised populations, including waste pickers as a particularly vulnerable group, should be improved. This may include, but are not limited to, policies to make healthy food options more affordable with consideration of a subsidy for the vulnerable, increased targeted social protection including food relief, and an extension of the health promotion levy to tax other unhealthy food items.

### 4.1. Significance for Public Health

Waste pickers are an extremely vulnerable population from an occupational health perspective [31]. In addition, their precarious income generation and fierce relative competition for “spots” on landfill sites to seek out a livelihood compromise their health seeking behaviour, yet they contribute to reduction of waste to landfill sites and environmental sustainability. Knowledge and understanding of the lived realities of waste pickers should guide service delivery planning by community nutrition and public health practitioners. Targeted service delivery to this vulnerable population could include the provision of health screening, primary health care and chronic care through mobile outreach services. In addition, a coordinated approach to extend service delivery to include aspects of social services will go a long way to improve their lives. These extended services could include applying for identity documents to allow the opportunity to access social grants, and registration on the indigent programme for free basic services such as water and electricity. The rendering of services to this population should be planned and executed with sensitivity and understanding of their unique working (and often also living) conditions. Waste pickers cannot afford the opportunity cost of not working for a day.

### 4.2. Limitations

Reported dietary intake is subject to bias caused by systematic underreporting, overreporting or omission of foods by an individual during the interviewing process. The convenience sample approach, including all participants available on the day and willing to participate, may introduce sample selection bias. It was not feasible to repeat the dietary intake assessment as the informal and fluid nature of the setting made it difficult to locate the respondents. The findings from this study are not regarded as representative of other settings.

## Figures and Tables

**Figure 1 nutrients-14-01172-f001:**
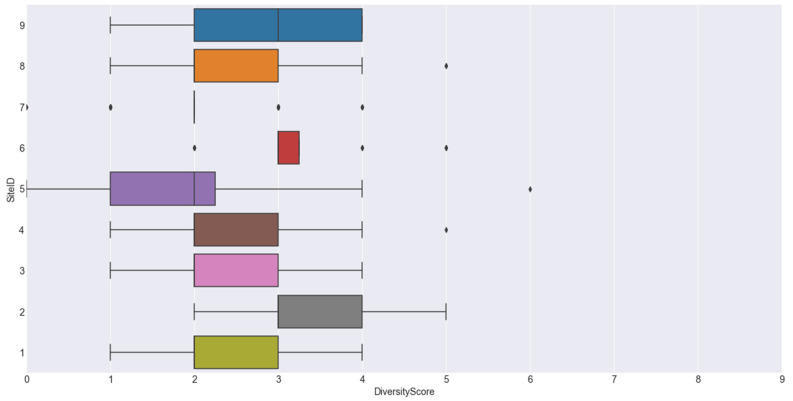
Box plot of dietary diversity scores per site. Sites 1, 3 and 7 were in urban areas. All other sites were in rural areas.

**Table 1 nutrients-14-01172-t001:** Comparison of BMI of waste pickers with 2016 SADHS* findings.

	Sex	Underweight			Overweight/Obese		
		Severely Thin ≤17	Mildly Thin (>17<18.5)	Total Thin (≤18.5)	Total Overweight (≥25)	Overweight (>25<30)	Obese (≥30)
Waste pickers 2015/16 (Chi-square *p* < 0.001)	Male (*n* = 110)	10 (4.6%)	26 (11.9%)	36 (16.5%)	19 (8.7%)	16 (7.3%)	3 (1.4%)
	Female (*n* = 190)	3 (1.8%)	8 (4.8%)	11 (6.7%)	84 (50.9%)	38 (23.0%)	46 (27.9%)
	TOTAL (*n* = 300)	13 (3.4%)	34 (8.9%)	47 (12.2%)	103 (26.9%)	54 (14.1%)	49 (12.8%)
SADHS 2016 *							
Lowest wealth quantile #	Male	2.9%	7.0%	9.9%	17.4%	14.1%	3.3%
Total		2.1%	7.4%	9.5%	31.3%	20.3%	11.0%
Lowest wealth quantile #	Female	0.6%	2.4%	3.0%	57.3%	27.8%	29.5%
Total		0.5%	2.1%	2.6%	67.7%	26.6%	41.0%

* South African Demographic and Health Survey, 2016 [26]. # The wealth quantile was based on a wealth index that scores households on the number and kinds of consumer goods owned using principal component analyses. Lowest wealth quantile represents 20% of SADHS 2016 [26] population that has fewest assets including those of lowest value.

**Table 2 nutrients-14-01172-t002:** Mean intakes of energy, macronutrients, added sugar and total fibre.

Gender		Energy (kJ)	Total Protein (g)	Total Fat (g)	Total Carbohydrate (g)	Added Sugar (g)	Total Fibre (g)
Male (*n* = 205)	Mean	7408.3	61.6	44.85	248.5	44.1	17.7
	SD	4651.8	39.8	38.8	155.5	63.9	12.6
	Median	6809.1	54.6	34.2	233.9	24.0	15.8
	Min	110.4	0.18	0	6.0	0	0
	Max	26,147.0	203.8	225.6	961.9	445.8	92.3
Female (*n* = 153)	Mean	5664.1	43.5	33.8	210.6	36.0	16.4
	SD	3583.8	28.8	29.7	132.4	52.3	12.0
	Median	4917.2	39.12	27.2	184.0	18.0	13.0
	Min	213.6	0	0	10.4	0	0
	Max	23,637.1	154.0	165.0	707.1	330.0	67.0
Total (*n* = 358)	Mean	6662.9	53.8	40.0	232.3	40.7	17.2
	SD	4310.5	36.6	35.5	147.1	59.3	12.3
	Median	6017.2	46.5	29.2	208.9	20.4	14.7
	Min	110.4	0	0	6.0	0	0
	Max	26,147.0	203.8	225.6	961.9	445.8	92.3
Proportional adequacy			Total protein *	Total fat *	Total Carbohydrate *		Fibre *
Inadequate (%)			23.4	50.0	1.0		100
Adequate (%)			76.0	36.6	47.5		0
Excessive (%)			0.6	13.4	51.6		0

* Compared to acceptable macronutrient distribution ranges for adults [27].

**Table 3 nutrients-14-01172-t003:** Summary of micronutrient consumption, compared with recommended intake.

	Males (*n* = 205)			Females (*n* = 153)		
	DRI *	Mean	% of DRI	DRI *	Mean	% of DRI
Calcium (mg)	800	286.88	35.9	800	257.52	32.2
Iron (mg)	6	14.35	239.2	5–8.1	11.85	148.1–237.0
Magnesium (mg)	330–350	272.52	77.9–82.6	255–265	210.51	79.4–82.6
Phosphorus (mg)	580	842.39	145.2	580	598.21	103.1
Potassium (mg)	4700	1547.17	32.9	4700	1227.84	26.1
Sodium (mg)	1500	1463.90	97.6	1500	883.4	58.9
Manganese (mg)	2.3	2.43	105.7	1.8	2.28	126.7
Zinc (mg)	9.4	11.88	126.4	6.8	8.66	127.4
Vitamin A RE (µg)	625	527.17	84.3	500	503.58	100.7
Thiamin (mg)	1.0	1.62	162.0	0.9	1.43	158.9
Riboflavin (mg)	1.1	1.12	101.8	0.9	0.90	100.0
Niacin (mg)	12	20.69	172.4	11	12.52	113.8
Vitamin B6 (mg)	1.1	3.52	320	1.1	2.27	206.4
Folate (µg)	320	365.67	114.2	320	333.62	104.3
Vitamin B12 (µg)	2.0	4.19	209.5	2.0	2.83	141.5
Pathothenate (mg)	5	3.40	68.0	5	2.24	44.8
Biotin (µg)	30	28.26	94.2	30	25.92	86.4
Vitamin C (mg)	75	17.77	23.7	60	20.18	33.6
Vitamin D (µg)	10	2.47	24.7	10	1.49	14.9
Vitamin E (mg)	12	6.39	53.3	12	6.43	53.6

* Estimated adequate requirement (EAR), where established, or adequate intake (AI) [28].

**Table 4 nutrients-14-01172-t004:** Summary of types of food consumed by waste pickers, according to food groups.

Food Groups	Types of Foods Reported to be Consumed by Waste Pickers
Cereals	Bread, breakfast cereal, hot cross bun, maize porridge, oats porridge, pasta, pizza, rice, samp (split white corn), scone, sorghum porridge, vetkoek (fried dough), Weet-Bix, wholegrain breakfast cereal
White roots and tubers	Potato, sweet potato
Vitamin A rich vegetables and tubers	Carrot, pumpkin
Vitamin A rich fruits	None
Dark green leafy vegetables	Amaranth leaves, broccoli, cabbage, Swiss chard
Other vegetables	Beetroot, garlic, lettuce, mixed vegetables, onion, sweet pepper, tomato
Other fruits	Apple, banana, grape, lemon, pear
Organ meat	Beef liver, chicken feet, chicken giblets, chicken head, sheep liver
Flesh meats	Bacon, beef, beef patty, beef sausage, chicken, mutton, ostrich, polony, pork, salami, turkey, vienna sausage
Fish and seafood	Fish, pilchards
Eggs	Egg
Legumes, nuts and seeds	Beans, peanuts, soya mince
Milk and milk products	Cheese, cheese spread, milk, yogurt
Oils and fats	Canola oil, margarine, mayonnaise, nondairy creamer, peanut butter, sunflower oil
Sweets	Carbonated cold drink, chocolate, chocolate coated bar, cold drink squash, condensed milk, dairy fruit juice mix, jam, glucose drink, sweets, sweetened orange juice, sugar
Spices, condiments and beverages	BBQ sauce, beer, Bovril (meat extract paste), coffee, curry sauce, fruit chutney, gravy, instant soup, mango achar, Rooibos tea, sorghum beer, soup powder, spirits, tomato sauce, tea, vinegar, wine

## Data Availability

The anonymised datasets used and analysed during the current study are available from the corresponding author on reasonable request.

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
