# Peer review of "Nutritional Status, Dietary Intake and Dietary Diversity of Landfill Waste Pickers"

_nutrients, 2022, doi:10.3390/nu14061172_

Round 1

Reviewer 1 Report

The current paper is an interesting study to show how COVID-19 has impacted the nutritional status and intake of vulnerable populations. The objectives, research questions and methodology and results sections will need to be clarified and improved as outlined in the detailed comments below.

Introduction

P2 line 53: What do you mean with limited access to food? Is it food in general or nutritious foods?

P2 lines 77-80: “In South Africa, there are an estimated 60,000 to 90,000 informal waste pickers operating at different levels (18). These individuals operate at the lowest level in the hierarchy of entities that collect 80 and dispose of waste, “ – What do you mean with the different levels, and to whom are you referring to with ‘these waste pickers operating at the lowest level’ is that the population of waste pickers that you included in your study?

P2 line 92: It would be helpful if you could also translate the poverty line to US dollars or Euros for more international reference.

P3 end of introduction: can you please specify the objectives and research question of the study?

Materials and methods:

Did you also collect socio-demographic information? Can you add the names of the cities/areas where the waste pickers were located?

Were the 24 h recall method and toolkit used validated?

How did you calculate the sample size?

Findings were compared to SADHS 2016, but a description on the comparison is missing in the methodology and should be added. It would be good to add a reference to SADHS 2016.

Results:

May be good to provide a description on the socio-demographics of the sample size, for example age, gender and sites

Table 1: can you add total numbers for sample sizes in a separate column and explain what SADHS is in a footnote?

Table 2: I would suggest to add respective dietary reference values and % of the population with inadequate intake or excessive intake where appropriate. You also will need to add these comparisons to the methods and add references for the dietary reference values used.

Table 3: It is incorrect to assess nutrient adequacy in a population by benchmarking the intake to DRI of the micronutrients, this should be the estimated average requirement (EAR). Can you also clarify the reference values for iron with respect to the bioavailability of the diet?

Did you assess zinc intake from the diet?

3.3. Dietary diversity scores: it would be helpful to add information to interpret the DDS and to translate that to the risk of nutrient inadequacy.

Were there any differences between genders in DDS?

Figure 1: this needs more clarification on the sites and whether they were urban or rural.

Discussion

Line 250-274: This information is also mentioned in the introduction and should not be repeated. Rather start the introduction with a summary of the main findings.

Line 275-280: The current paper will need a summary of the descriptives in the results section, with a reference to the article where you describe them in more detail. If you want to refer to income then please put it into relation with the nutritional findings.

Line 364-366: The significant variation in dietary intake among waste pickers in this study is likely an indication of varied access to food retrieved from the landfill site. – It unclear why you make this statement when the DDS is relatively low.

The discussion would benefit from a paragraph on strengths and limitations of the study and the implications for the conclusions

A clear conclusion at the end of the discussion is missing.

Reviewer 2 Report

Thank you for the opportunity to review the article “Nutritional Status, Dietary Intake and Dietary Diversity of Landfill Waste Pickers.” The article is interesting and discusses the nutritional status, dietary intake and dietary diversity of waste pickers in South Africa.

Major comments:

  1. Table 4 in line 192 should be ‘supplemental table 4’.
  2. In line 207, ‘processed meats were consumed by 14.4% of the participants’ should be supplemental table 5.

Minor comments:

  1. Adding a table summarizing food consumption based on food group in the manuscript will improve the clarity of the manuscript.
  2. In Table 1, is the SADHS 2016 statistics for the whole South African population?
  3. In the ‘limitations’ section, the authors need to also mention other biases such as sample selection bias.
